# Dietary Patterns and Cardiovascular Risk Factors in Spanish Adolescents: A Cross-Sectional Analysis of the SI! Program for Health Promotion in Secondary Schools

**DOI:** 10.3390/nu11102297

**Published:** 2019-09-26

**Authors:** Patricia Bodega, Juan Miguel Fernández-Alvira, Gloria Santos-Beneit, Amaya de Cos-Gandoy, Rodrigo Fernández-Jiménez, Luis Alberto Moreno, Mercedes de Miguel, Vanesa Carral, Xavier Orrit, Isabel Carvajal, Carolina E. Storniolo, Anna Tresserra-Rimbau, Mónica Doménech, Ramón Estruch, Rosa María Lamuela-Raventós, Valentín Fuster

**Affiliations:** 1Foundation for Science, Health and Education (SHE), 08008 Barcelona, Spain; gsantos@fundacionshe.org (G.S.-B.); adecos@fundacionshe.org (A.d.C.-G.); mdemiguel@fundacionshe.org (M.d.M.); vcarral@fundacionshe.org (V.C.); xorrit@fundacionshe.org (X.O.); icarvajal@fundacionshe.org (I.C.); 2Centro Nacional de Investigaciones Cardiovasculares (CNIC), 28029 Madrid, Spain; jmfernandeza@cnic.es (J.M.F.-A.); rodrigo.fernandez@cnic.es (R.F.-J.); 3The Zena and Michael A. Wiener Cardiovascular Institute, Icahn School of Medicine at Mount Sinai, New York, NY 10029, USA; 4Centro de Investigación Biomédica En Red en enfermedades CardioVasculares (CIBERCV), 28029 Madrid, Spain; 5GENUD (Growth, Exercise, NUtrition and Development) Research Group, Faculty of Health Science, University of Zaragoza, 50009 Zaragoza, Spain; lmoreno@unizar.es; 6Consorcio CIBER, M.P. Fisiopatología de la Obesidad y Nutrición (CIBERObn), Instituto de Salud Carlos III (ISCIII), 28029 Madrid, Spain; carolinastorniolo@outlook.com (C.E.S.); anna.tresserra@iispv.cat (A.T.-R.); MDOMEN@clinic.cat (M.D.); RESTRUCH@clinic.cat (R.E.); lamuela@ub.edu (R.M.L.-R.); 7Department of Nutrition, Food Science and Gastronomy, School of Pharmacy and Food Sciences, XaRTA, INSA, University of Barcelona, 08028 Barcelona, Spain; 8Universitat Rovira i Virgili, Departament de Bioquímica i Biotecnologia, Unitat de Nutrició Humana. Hospital Universitari Sant Joan de Reus, Institut d’Investigació Pere Virgili (IISPV), 43201 Reus, Spain; 9Department of Internal Medicine, Hospital Clínic, Institut d’Investigacions Biomèdiques August Pi I Sunyer (IDIBAPS), University of Barcelona, 08036 Barcelona, Spain

**Keywords:** dietary patterns, adolescents, cardiovascular health, cluster analysis, principal components analysis

## Abstract

Previous studies on the association between dietary habits and cardiovascular risk factors (CVRF) in adolescents have generated conflicting results. The aim of this study was to describe dietary patterns (DP) in a large sample of Spanish adolescents and to assess their cross-sectional relationship with CVRF. In total, 1324 adolescents aged 12.5 ± 0.4 years (51.6% boys) from 24 secondary schools completed a self-reported food frequency questionnaire. DPs were derived by cluster analysis and principal component analysis (PCA). Anthropometric measurements, blood pressure, lipid profile, and glucose levels were assessed. Linear mixed models were applied to estimate the association between DPs and CVRF. Three DP-related clusters were obtained: *Processed* (29.2%); *Traditional* (39.1%); and *Healthy* (31.7%). Analogous patterns were obtained in the PCA. No overall differences in CVRF were observed between clusters except for z-BMI values, total cholesterol, and non-HDL cholesterol, with the *Processed* cluster showing the lowest mean values. However, differences were small. In conclusion, the overall association between DPs, as assessed by two different methods, and most analyzed CVRF was weak and not clinically relevant in a large sample of adolescents. Prospective analysis may help to disentangle the direction of these associations.

## 1. Introduction

Cardiovascular (CV) diseases have a multifactorial origin, and modifiable behavioral risk factors such as insufficient physical activity, unhealthy diet, or smoking play an essential role in disease development and progression. Suboptimal diet is one of the leading risk factors for death and disability in the world [1,2].

Adult behavior has its roots in childhood [3]. Adolescence is a critical period of personal development when individuals acquire greater autonomy [4]. This period is characterized by biological, psychological, and social changes that expose the adolescent to unhealthy behaviors [5], and consequently the prevalence of unhealthy behaviors and risk factors in adolescents is alarmingly high [6]. The consumption of low nutritional quality diets is especially prevalent in groups with lower socioeconomic status [7,8]. Nevertheless, according to the results of the HELENA study, the proportion of European adolescents achieving the American Heart Association (AHA) ideal diet score is as low as 2% [9].

Previous reports have described the main dietary patterns (DP) in adolescent populations and their associations with several health outcomes [10,11,12]. However, results are inconsistent, with some studies showing positive associations between healthier DPs and better CV health profiles, while others have shown the opposite pattern. Information provided by these studies is also useful for improving current dietary guidelines [4,13]. The methods most frequently used to identify DPs are cluster analysis and principal component analysis (PCA). These methods differ in important respects: While cluster analysis classifies individuals into mutually exclusive groups or clusters with similar food intake [4], PCA uses the correlation between different food groups, identifying linear combinations of foods that are frequently consumed together [14]. These differences might limit comparability of results across studies using cluster analysis vs. those using PCA. To date, very few studies have analyzed DPs using both methods. Testing the comparability of cluster analysis and PCA in adolescent populations will thus help to clarify whether studies using these approaches are analogous.

The aim of this study was to define DPs using both cluster analysis and PCA and to assess their associations with CV risk factors (CVRF) in a large sample of Spanish adolescents enrolled in the SI! Program for Secondary School [15].

## 2. Materials and Methods

The SI! Program (*Salud Integral*—Comprehensive Health) (https://www.programasi.org/) is a school-based multicomponent educational intervention aimed at promoting CV health in childhood and adolescence [15,16,17,18,19]. The SI! Program for Secondary school enrolled 1326 adolescents and their families in 2017 from 24 public high schools in Madrid and Barcelona, Spain. The overall rationale and study design, as well as the protocol for data collection and measurements, have been previously described [15], and is registered at ClinicalTrials.gov, number NCT03504059. Collected data at study baseline were used for the analyses. The study was approved by the ethical committees of the institutions involved in the project (University of Barcelona, *Centro Nacional de Investigaciones Cardiovasculares* (CNIC), and the SHE Foundation). Participants provided informed consent for all examinations.

### 2.1. Dietary Assessment

Dietary data were obtained using the Children’s Eating Habits Questionnaire (CEHQ-FFQ), which is a validated food frequency questionnaire (FFQ) [20,21]. The CEHQ-FFQ assesses the consumption of 43 food items during the preceding four weeks. Response options displayed from left to right were as follows: ‘Never/less than once a week’, ‘1–3 times a week’, ‘4–6 times a week’, ‘1 time per day’, ‘2 times per day’, ‘3 times per day’, ‘4 or more times per day’, and ‘I have no idea’ [21]. The CEHQ-FFQ does not provide information about total energy or food intake; instead, it focuses on the main food groups that have been shown to be related to childhood overweight and overall health [20,21]. Adolescents self-reported their frequencies of consumption using an online application under the guidance of trained nutritionists. For the analysis, a conversion factor ranging from 0 to 30 was used to transform questionnaire answers into weekly consumption frequencies [22].

### 2.2. Anthropometric and Clinical Measures

Body weight was measured with an OMRON BF511 electronic scale and height with a Seca 213 portable stadiometer, with adolescents wearing light clothes and no shoes. Body mass index (BMI) was calculated as body weight divided by height squared (kg/m^2^). Waist circumference (WC) was measured in triplicate with a Holtain tape to the nearest 0.1 cm. Total fat content was estimated by bioelectrical impedance analysis with a tetra polar OMRON BF511 device. Fat mass index (FMI) was calculated by dividing body fat mass (kg) by height squared (m^2^). Blood pressure (BP) was measured with an OMRON M6 monitor three times at 2–3 min intervals according to a standardized protocol [23], and the minimum reading was used for analysis. Age- and sex-adjusted BMI, BP, and WC percentiles and z-scores were calculated according to Center for Disease Control standards [24], the American Academy of Pediatrics [25], and NHANES III [26], respectively. Sex-specific FMI z-scores were calculated based on our sample. Maturation stages according to Tanner [27] were self-reported by participants with the help of pictograms. Blood glucose levels and lipid profiles (total cholesterol (TC), HDL-cholesterol, non-HDL-cholesterol, LDL-cholesterol, triglycerides (TG), and TC:HDL and LDL:HDL ratios) were analyzed using the CardioCheck Plus device in capillary blood sampled with a lancet [28]. All measurements were carried out by trained nutritionists and nurses.

### 2.3. Physical Activity

Physical activity levels were estimated by the triaxial Actigraph wGT3X-BT accelerometer, worn on the non-dominant wrist for 7 consecutive days. Records were considered valid if they provided data from a minimum of 4 days with at least 600 minutes of wear time. Chandler (2016) [29] cut-off points were applied for the calculation of time spent in different physical activity intensities. Time spent in moderate-to-vigorous physical activity was used as a covariate in the main models.

### 2.4. Family Information

Parents filled in a self-administered questionnaire on family socioeconomic status. Parental educational level was categorized according to the International Standard Classification of Education (ISCED) [30]: Low (families without studies or primary studies, 0 to 1 ISCED level), medium (secondary studies and professional training, 2 to 4 ISCED levels), and high (university studies, 5 to 6 ISCED level). For the analyses, the highest education level of either parent was taken into account. A migrant background was assumed if at least one of the parents was born outside Spain.

### 2.5. Statistical Methods

Continuous variables were expressed as mean and standard deviation or 95% confidence intervals, whereas categorical variables were expressed as frequency distributions and percentages.

DP clusters were identified by a combined methodology including hierarchical (Ward’s method) and non-hierarchical (k-means) approaches, as previously described [31]. Relative consumption frequency was calculated for each food item by dividing the consumption frequency of the specific food item by the sum of the consumption frequencies of all food items reported by each individual participant. Food items were standardized before clustering in order to avoid differences in variances that might affect the clustering results. Ward’s method based on squared Euclidean distances was applied, generating and comparing several cluster solutions from two to five clusters. With the resulting centroids, a K-means cluster analysis algorithm was applied with a predefined maximum 100 iterations to generate a fine-tuned clustering solution. The stability of the final solution was examined by randomly splitting the database into halves and repeating the same clustering procedure and comparing both solutions, until satisfactory results were observed (kappa degree of concordance = 0.91 and 0.88 for the first and second halves, respectively).

DPs were also derived by PCA using the same standardized relative consumption frequencies. The number of factors to be retained was mainly determined by the scree plot and the interpretability of the DPs [32]. Interpretability was improved by varimax (orthogonal) rotation. Food items with absolute factor loadings greater than 0.30 were considered important contributors to a DP. Labels were applied to clusters based on foods with higher consumption frequencies and to principal components based on foods with higher loadings within components. To compare the results of the two methods, mean principal component scores were calculated for each cluster.

Multilevel linear mixed-effect models were used to assess the association between DPs and CVRF (anthropometric measurements, BP, blood glucose, and lipid profile). Fixed effects were age (continuous variable), gender and fasting status (binary variables), z-BMI (continuous variable), moderate-to-vigorous physical activity (minutes, continuous variable), and migrant background and parental educational level (categorical variables). Schools and municipality (Madrid or Barcelona) were handled as random effects. In the models evaluating z-BMI, FMI, and WC as dependent variables, the Tanner stage (categorical variable, from I to V) was also included as a fixed effect. Statistical significance was set at a threshold of 0.05. Bonferroni correction was used to adjust for multiple pairwise comparisons. All analyses were performed using SPSS Statistics for Windows, version 19.0.

## 3. Results

A total of 1324 adolescents (of 1326 enrolled) completed the FFQs at baseline. Mean age of adolescents was 12.5 ± 0.4 years (51.6% boys), and participants were enrolled at 24 Secondary Schools in Barcelona and Madrid (Spain). Based on 43 food items and their relative consumption frequency, categorization into three clusters were considered the most interpretable and stable solution. The three identified clusters were labelled as follows: *Processed* (*n* = 386, 29.2%), *Traditional* (*n* = 518, 39.1%), and *Healthy* (*n* = 420, 31.7%). The mean z-score and standard deviation for all food items in the three clusters are presented in Table 1.

Adolescents in the *Processed* cluster more frequently consumed crisps, popcorn, sweetened drinks, hamburgers, hot dogs, ice cream, and sauces like ketchup and mayonnaise, whereas their consumption of unsweetened cereals, meat and non-fried fish, vegetables, fruit with no added sugar, and white bread was less frequent than in the other clusters. Compared with the other clusters, participants in the *Traditional* cluster had higher relative consumption frequencies of water, white bread, pasta and rice, sweetened milk and yogurt, cold cuts, and ready-to-cook meat products and lower scores for unsweetened cereals, boiled eggs, and vegetables. The highest scoring food items in the *Healthy* cluster were fruit with no added sugar, vegetables, unsweetened cereals, and unsweetened milk and yogurt, while this cluster had relatively low scores for fried potatoes, sauces, sweetened drinks and milk, and fast foods such as pizza and hamburgers.

PCA of the same variables also yielded three DPs, explaining 30.4% of the variation in the sample (Table 2). The *Processed* component was characterized by high loadings on high-sugar and high-fat foods such as chocolate and candy bars, pizza, crisps, popcorn, candies, and ice cream. The *Traditional* component featured high-factor loadings on fresh and fried fish and meat, cold cuts, pasta, rice, and sweetened milk. The *Healthy* component was characterized by high loadings on vegetables, fruit with no added sugar, and unsweetened cereals.

Comparison of the mean principal component values across the derived DP clusters is shown in Figure 1. The *Processed* component score was significantly higher in the *Processed* cluster than in the *Healthy* and *Traditional* clusters (*p* < 0.001), whereas the *Traditional* component score was significantly higher in the *Traditional* cluster than in the *Healthy* cluster (*p* < 0.001) but not the *Processed* cluster (*p* = 0.089). The *Healthy* component score was significantly higher in the *Healthy* cluster than in the *Traditional* and *Processed* clusters (*p* < 0.001).

The general characteristics of adolescents according to DP clustering are presented in Table 3. The proportion of girls in the *Healthy* cluster (55.7%) was significantly higher than in the other clusters. The *Healthy* cluster was also the most prevalent among participants whose parents had a high educational level; in contrast, the *Processed* cluster contained higher proportions of participants from families with a low or medium parental educational level (26.2% and 43.8%, respectively). The *Processed* cluster also contained the highest proportion of participants from a migrant background (37.8%).

The associations between cluster-analysis–derived DPs and CVRF are shown in Table 4. Mean z-BMI was significantly higher in the *Traditional* (mean 0.46, 95% CI: 0.35, 0.57) and *Healthy* (mean 0.44, 95% CI: 0.32, 0.57) clusters than in the *Processed* cluster (mean 0.11, 95% CI: −0.02, 0.24). Mean z-FMI was significantly higher in the Traditional cluster (mean 0.03, 95% CI: −0.03, 0.08) than in the Processed cluster (mean −0.06, 95% CI: −0.12, 0.00). TC was significantly higher in the *Healthy* cluster (mean 157.6 mg/dL, 95% CI: 150.7, 164.4) than in the *Processed* cluster (mean 148.6 mg/dL, 95% CI: 141.6, 155.5). This variation was reflected in the non-HDL cholesterol differences between the *Healthy* cluster (mean 95.1 mg/dL, 95% CI: 90.8, 99.5) and the *Processed* cluster (mean 89.0 mg/dL, 95% CI: 84.4, 93.6). No statistically significant differences were found between clusters for the remaining CVRF analyzed. Analysis of the association of PCA-derived DPs with CVRF showed similar results to those found by cluster analysis (Appendix A
Table A1).

## 4. Discussion

Three DPs (*Processed*, *Traditional*, and *Healthy*) were derived by cluster analysis and PCA of the food consumption frequencies in a large sample of adolescents participating in the SI! Program for Secondary School in Spain. The similarity of the DPs derived by the two methods emphasizes the robustness of the results obtained. Because DPs are sensitive to the specific profile of the population studied, it is difficult to compare results between studies; however, similar DPs have been reported in other populations [14,31,33,34,35]. In seven-year-old children, Northstone et al. described a *Processed* DP characterized by the consumption of processed meat, coated poultry and fish products, chips, pizza, and sandwiches [33]. Likewise, Aranceta et al. defined a *Snacky* pattern (in a population aged from 2 to 24 years) characterized by high consumption of bakery products, sweets, salted snacks, and soft drinks [34]. Healthy patterns have been also frequently described. Pérez-Rodrigo et al. described a *Mediterranean DP* characterized by high consumption of vegetables, olive oil, fish, fruit, yogurt, and water in Spanish population aged 2 to 24 years [31]. The traditional patterns described in the literature reflect country-specific dietary habits that are especially difficult to compare between studies. For example, Smith et al. characterized a *Traditional British* cluster rich in full-fat milk, meat, and potatoes [14], while Lee et al. described a *Korean Traditional* pattern with high consumptions of grains, kimchi, fish and shellfish, beef, vegetables, seaweeds, oils, and oriental sauces [35].

Our analysis of associations between DPs and CVRF only found significant associations for z-BMI, TC, and non-HDL cholesterol, with z-BMI in particular being higher in the *Traditional* and *Healthy* clusters than in the *Processed* cluster. The literature on the association between DPs and CV risk and health factors shows conflicting results. Some studies have shown an association between high healthy DP scores and favorable biomarkers, a better CV profile, and a lower risk of being overweight [10,11,31,36,37]. However, other studies have shown opposite results [12,35,36,38,39]. A cross-sectional study in 12–16-year-old American adolescents found inverse associations between the prevalence of excess central body fat and the intakes of dairy products, whole grains, total fruit, and vegetables [38]. Truthmann et al. found that adolescents with a good quality diet (assessed by a dietary index) had a higher rate of obesity and higher BP than those with a poorer diet [36]. Likewise, data from the 2001 South Korea National Health and Nutrition Survey [35] showed that children and adolescents allocated to a *Traditional diet* group had higher rates of obesity than participants in the *Westernized fast-food* and *Mixed-diet* clusters. The same study found the highest levels of TG and the lowest levels of TC and HDL-cholesterol in participants allocated to the *Traditional* cluster, although these differences were not statistically significant. Surprisingly, the highest levels of HDL-cholesterol were found in participants belonging to the *Westernized fast-food* cluster [35]. Another study showed a similar pattern, with TC higher among participants with the *Healthy* DP than among those in the *Unhealthy* DP [12]. In another study, Golley et al. found no associations between dietary quality and serum lipid concentration [40].

The mechanisms underlying these associations are poorly understood. One major concern is participant under-reporting or over-reporting of intake, especially in relation to high-fat foods [39,41]. Other factors that might affect long-term body composition are the method of preparation (e.g., deep frying vs. stir frying) and serving style (e.g., with sauce or side dishes) [38,42,43]. Our study did not quantify total energy intake, and the results are based on frequency of food intake. Similar consumption frequencies thus might mask differences in portion size, preparation, and serving style that result in differing total calorie content. This might be especially relevant in the context of socioeconomic differences. In our study, adolescents from families reporting a relatively lower socioeconomic status were more frequently allocated to the *Processed* pattern but still could be consuming less total energy than required due to food insecurity, and thus confounding the associations with clinical markers. The participants in our study are also facing the onset of puberty, which may influence dietary habits, food intake reporting, and biomarker values. Adolescents are known to sometimes reduce their food intake or misreport consumption because of weight concerns [36]. Another possibility is that the lower BMI, FMI and TC in the *Processed* pattern might be a product of reverse causation, with participants with a high BMI, FMI or a worse cholesterol profile altering their food intake toward a healthier profile. Participants with a higher BMI/FMI might also be more susceptible to social desirability bias. Even if they reflect a genuine association, differences in body composition and biomarkers according to DPs are unlikely to be clinically relevant because all mean values remained within the normal range and differences were small. Future prospective analyses will provide insight into the long-term trends in body composition and CV health associated with DPs, clarifying the true direction of the associations.

Although the benefits of a healthy diet are widely known, fruit and vegetable intake is influenced by taste preferences, repeated exposure, social experience, and availability [44]. The value of derived DPs is that this approach considers diet as a whole rather than a collection of nutrients or components [14]. Cluster analysis classifies individuals into non-overlapping groups or clusters based on similarities between diets, while PCA exploits the correlation between different food groups by identifying linear combinations of foods that are frequently consumed together. Both approaches describe diet as a whole, but from slightly different perspectives, despite the aim of both methods being to explain maximum variation in food intake [4,14]. It is common practice to use just one of these approaches; however, we decided to use both methods to assess the consistency of the associations between the derived DPs and CVRF. The food items forming each DP were mostly the same, independently of the method used. The best agreement was found for the *Healthy* patterns, in which the 10 items showing the highest frequencies of consumption were the same in both approaches. Conversely, the lowest agreement was found for the *Traditional* patterns; they tended to correlate with the *Processed* patterns and showed weaker associations with clinical markers, making their interpretation more difficult.

A similar agreement between cluster and PCA analyses was found in two previous studies. In seven-year-old children from the ALSPAC cohort, Smith et al. derived three similar DPs by cluster analysis (*Processed*, *Plant-based*, and *Traditional British*) and PCA (*Junk*, *Traditional*, and *Health-conscious*) [14]. In another study on Irish adolescents aged 13–17 years from the Irish National Teens Food Survey, the DPs derived by cluster analysis were *Healthy*, *Unhealthy*, *Rice/Pasta dishes*, *Sandwich*, and *Breakfast cereal and Main meal-type foods*, whereas the PCA-derived DPs were *Healthy foods*, *Traditional foods*, *Sandwich foods*, and *Unhealthy foods*. Although the two methods identified different numbers of DPs, the authors concluded that the results were still comparable [45].

This study has several strengths. The SI! Program for Secondary School is a large multi-center study in which all nurses and nutritionists received prior training according to strictly standardized procedures for data collection and guiding participants in completing the questionnaire. Another strength is the use and comparison of two different methods to derive DPs, thus providing more robust evidence and consistency in the results found. Moreover, the study focuses on pre-adolescents, who receive less attention in lifestyle studies than younger children or older adolescents. Finally, follow-up of these adolescents will provide the opportunity to track changes in DPs and to assess their association with changes in CV health over time. Some study limitations should be acknowledged. The cross-sectional nature of the analyses allows the establishment of associations but not causation. Participation in the study was voluntary, and we therefore cannot exclude selection bias due to adolescents with worse CV health or unhealthier lifestyles being less willing to participate. Another limitation is the self-reported nature of the CEHQ-FFQ, which introduces the possibility of social desirability bias or difficulties in the accurate estimation of intake frequency for certain food groups.

## 5. Conclusions

Cluster analysis and PCA identified three consistent DPs (*Processed*, *Traditional*, and *Healthy*) in a large sample of Spanish adolescents. Overall, no significant associations were found between DPs and CVRF, with the exception of z-BMI, z-FMI, TC, and non-HDL cholesterol. Given the limitations of the cross-sectional design (including the potential for reverse causation in the association between DPs and CVRF), disentangling the direction of the associations between DPs and lipid profile and body composition in adolescents will require future prospective analyses.

## Figures and Tables

**Figure 1 nutrients-11-02297-f001:**
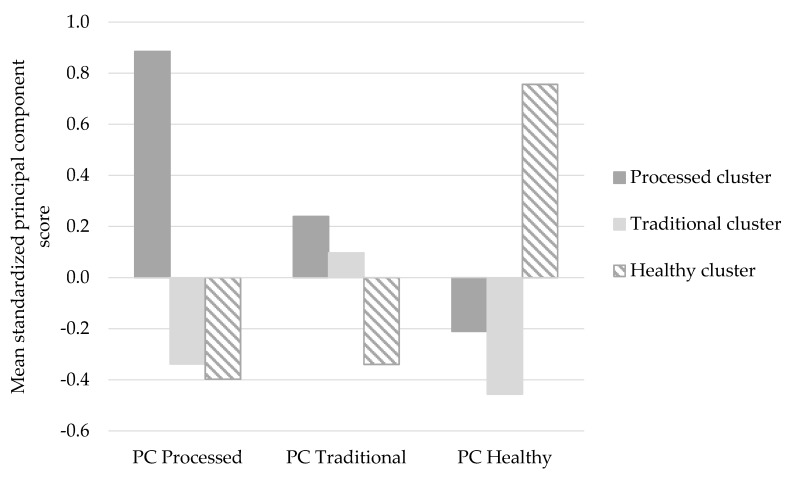
Mean standardized principal component scores for each cluster.

**Table 1 nutrients-11-02297-t001:** Z-Scores for relative consumption frequencies characterizing each cluster.

Food Items	Processed (*n* = 386)		Traditional (*n* = 518)		Healthy (*n* = 420)	
	Mean	SD	Mean	SD	Mean	SD
Cooked vegetables, potatoes, beans	−0.28	0.75	−0.23	0.55	**0.54**	1.36
Fried potatoes, potato croquettes	**0.42**	1.23	−0.08	0.78	−0.30	0.87
Raw vegetables	−0.32	0.67	−0.22	0.73	**0.56**	1.27
Fresh fruit with no added sugar	−0.49	0.57	−0.12	0.86	**0.60**	1.16
Fresh fruit with added sugar	**0.29**	1.15	−0.15	0.82	−0.08	1.00
Water	−0.72	0.85	**0.47**	0.86	0.09	0.90
Fruit juice	**0.33**	1.24	−0.12	0.86	−0.15	0.84
Sweetened drinks	**0.53**	1.43	−0.15	0.68	−0.30	0.58
Diet drinks	**0.38**	1.38	−0.18	0.55	−0.13	0.92
Breakfast cereals, muesli (sweetened)	**0.05**	0.99	0.00	1.00	−0.05	1.02
Porridge, oatmeal, cereals, muesli (unsweetened)	−0.09	0.78	−0.32	0.46	**0.48**	1.40
Plain unsweetened milk	−0.27	0.67	−0.19	0.82	**0.48**	1.26
Sweetened milk	−0.05	0.78	**0.35**	1.18	−0.39	0.77
Plain unsweetened yogurt or kefir	−0.05	0.88	−0.21	0.82	**0.30**	1.21
Sweet yogurt, fermented milk beverages	0.02	0.86	**0.15**	1.18	−0.21	0.84
Fresh or frozen fish, not fried	−0.28	0.97	−0.08	0.83	**0.36**	1.11
Fried fish, fish fingers	**0.17**	0.95	0.02	1.00	−0.18	1.02
Cold cuts, preserved, ready to cook meat products	−0.22	0.83	**0.31**	1.14	−0.18	0.85
Fresh meat, not fried	−0.20	0.98	0.01	0.91	**0.18**	1.09
Fried meat	**0.26**	1.14	0.06	0.97	−0.31	0.80
Fried or scrambled eggs	**0.25**	1.13	0.01	0.95	−0.24	0.87
Boiled or poached eggs	**0.22**	1.23	−0.23	0.73	0.09	0.99
Mayonnaise, mayonnaise-based products	**0.41**	1.28	−0.06	0.95	−0.30	0.53
Meat replacement products, plant milk	−0.06	0.52	−0.19	0.35	**0.29**	1.62
Cheese	**0.23**	1.25	−0.08	0.95	−0.11	0.74
Jam, honey	0.09	1.06	−0.20	0.72	**0.16**	1.19
Chocolate or nut-based spreads	**0.42**	1.20	−0.01	0.93	−0.37	0.69
Butter, margarine on bread	**0.27**	1.19	−0.16	0.86	−0.05	0.93
Reduced-fat products on bread	**0.24**	1.21	−0.18	0.77	0.00	0.99
Ketchup	**0.46**	1.28	−0.09	0.79	−0.31	0.75
White bread, white roll, white crispbread	−0.31	0.66	**0.43**	1.20	−0.24	0.78
Wholemeal bread, dark roll, dark crispbread	−0.08	0.85	−0.15	0.85	**0.26**	1.22
Pasta, noodles, rice	0.02	0.99	**0.02**	1.05	−0.05	0.95
Cereals, milled	**0.26**	1.47	−0.22	0.42	0.03	0.90
Pizza as main dish	**0.40**	1.23	−0.04	0.86	−0.32	0.78
Hamburgers, hot dogs, kebabs, wraps, falafel	**0.51**	1.29	−0.10	0.80	−0.34	0.69
Nuts, seeds, dried fruits	**0.22**	1.31	−0.20	0.73	0.04	0.91
Crisps, maize (corn) crisps, popcorn	**0.54**	1.34	−0.05	0.74	−0.44	0.61
Savory pastries, fritters	**0.44**	1.22	−0.15	0.80	−0.22	0.85
Chocolate, candy bars	**0.38**	1.28	−0.08	0.92	−0.25	0.64
Candies, loose candies, marshmallows	**0.43**	1.30	−0.10	0.83	−0.28	0.70
Biscuits, packaged cakes, pastries, puddings	**0.14**	0.91	0.13	1.19	−0.29	0.74
Ice cream, milk, or fruit-based bars	**0.47**	1.33	−0.17	0.79	−0.22	0.69

Values are expressed as mean with standard deviation. Bold entries are the highest mean value within a row.

**Table 2 nutrients-11-02297-t002:** Principal components and factor loadings.

Food Items	Processed	Traditional	Healthy
Cooked vegetables, potatoes, beans	0.07	0.14	**0.61**
Fried potatoes, potato croquettes	**0.57**	0.14	0.11
Raw vegetables	0.03	0.10	**0.57**
Fresh fruit with no added sugar	−0.03	−0.04	**0.58**
Fresh fruit with added sugar	**0.44**	0.24	0.20
Water	−0.12	0.09	−0.02
Fruit juice	**0.38**	0.21	0.14
Sweetened drinks	**0.52**	0.14	0.02
Diet drinks	**0.40**	0.11	0.17
Breakfast cereals, muesli (sweetened)	**0.32**	0.22	0.17
Porridge, oatmeal, cereals, muesli (unsweetened)	0.26	−0.04	**0.55**
Plain unsweetened milk	0.07	−0.08	**0.50**
Sweetened milk	0.25	**0.43**	−0.03
Plain unsweetened yogurt or kefir	0.10	0.10	0.43
Sweet yogurt, fermented milk beverages	0.29	**0.35**	0.07
Fresh or frozen fish, not fried	−0.03	0.15	**0.42**
Fried fish, fish fingers	0.20	**0.50**	0.19
Cold cuts, preserved, ready to cook meat products	0.02	**0.53**	0.04
Fresh meat, not fried	−0.03	**0.52**	**0.32**
Fried meat	0.15	**0.69**	0.11
Fried or scrambled eggs	**0.38**	**0.52**	0.16
Boiled or poached eggs	**0.42**	0.10	0.27
Mayonnaise, mayonnaise-based products	**0.36**	**0.52**	−0.05
Meat replacement products, plant milk	0.03	0.00	**0.31**
Cheese	**0.41**	**0.40**	0.18
Jam, honey	**0.35**	0.25	0.21
Chocolate or nut-based spreads	**0.63**	**0.36**	−0.04
Butter, margarine on bread	0.28	**0.41**	0.00
Reduced-fat products on bread	**0.42**	**0.33**	0.16
Ketchup	**0.43**	0.25	0.05
White bread, white roll, white crispbread	0.10	**0.32**	−0.05
Wholemeal bread, dark roll, dark crispbread	0.10	0.26	**0.34**
Pasta, noodles, rice	0.23	**0.49**	0.15
Cereals, milled	**0.31**	0.19	0.30
Pizza as main dish	**0.63**	0.11	0.03
Hamburgers, hot dogs, kebabs, wraps, falafel	**0.49**	**0.35**	0.06
Nuts, seeds, dried fruits	**0.37**	**0.33**	0.24
Crisps, maize (corn) crisps, popcorn	**0.63**	**0.34**	−0.08
Savouy pastries, fritters	**0.57**	0.23	0.06
Chocolate, candy bars	**0.65**	0.16	−0.02
Candies, loose candies, marshmallows	**0.60**	−0.04	0.01
Biscuits, packaged cakes, pastries, puddings	**0.53**	0.09	0.01
Ice cream, milk-, or fruit-based bars	**0.61**	0.19	0.11

Bold entries are factor loadings >0.30, which are considered important contributors to the patterns.

**Table 3 nutrients-11-02297-t003:** Adolescent and family characteristics in the total population and stratified by dietary pattern cluster.

	Total	Processed	Traditional	Healthy	*p*-Value
Age, years, mean (SD)	12.54 (0.45)	12.66 (0.55)	12.52 (0.40)	12.46 (0.37)	<0.001
Gender, *n* (%)					<0.001
Boys	683 (51.59%)	230 (59.59%)	267 (51.54%)	186 (44.29%)	
Girls	641 (48.41%)	156 (40.41%)	251 (48.46%)	234 (55.71%)	
Municipality, *n* (%)					0.302
Barcelona	901 (68.05%)	265 (68.65%)	362 (69.88%)	274 (65.24%)	
Madrid	423 (31.95%)	121 (31.35%)	156 (30.12%)	146 (34.76%)	
Parental education level, *n* (%)					<0.001
Low	214 (16.16%)	101 (26.17%)	68 (13.13%)	45 (10.71%)	
Medium	529 (39.95%)	169 (43.78%)	222 (42.86%)	138 (32.86%)	
High	518 (39.12%)	91 (23.58%)	203 (39.19%)	224 (53.33%)	
Unknown	63 (4.76%)	25 (6.48%)	25 (4.83%)	13 (3.10%)	
Parental origin, *n* (%)					<0.001
Spanish	850 (64.20%)	216 (55.96%)	371 (71.62%)	263 (62.62%)	
Migrant background	397 (29.98%)	146 (37.82%)	113 (21.81%)	138 (32.86%)	
Unknown	77 (5.82%)	24 (6.22%)	34 (6.56%)	19 (4.52%)	
MVPA, min/day, mean (SD)	76.63 (22.75)	76.47 (23.94)	76.56 (23.45)	76.55 (23.43)	0.996

MVPA: Moderate-to-vigorous physical activity. Values are expressed as mean (standard deviation) for continuous variables and as frequencies (percentages) for categorical variables. *p*-values from X^2^ and ANOVA test.

**Table 4 nutrients-11-02297-t004:** Cardiovascular risk factors according to dietary pattern clustering.

	*n*	Processed Cluster		Traditional Cluster		Healthy Cluster		*p* For Trend
		Mean *	95% CI	Mean *	95% CI	Mean *	95% CI	
*Anthropometry*								
z-BMI	1109	0.11 ^a,b^	−0.02, 0.24	0.46 ^a^	0.35, 0.57	0.44 ^b^	0.32, 0.57	<0.001
z-WC	1109	0.27	0.17, 0.36	0.26	0.17, 0.34	0.28	0.19, 0.37	0.719
z-Systolic BP	1106	−0.08	−0.26, 0.10	−0.12	−0.29, 0.05	−0.13	−0.31, 0.05	0.760
z-Diastolic BP	1106	−0.07	−0.24, 0.11	−0.04	−0.21, 0.13	−0.06	−0.24, 0.11	0.813
z-FMI	1109	−0.06	−0.12, 0.00	0.03	−0.03, 0.08	0.01	−0.05, 0.06	0.048
*Biochemistry*								
Glucose (mg/dL)	1109	104.2	101.4, 107.1	102.4	99.8, 105.1	103.5	100.7, 106.3	0.274
TC (mg/dL)	1109	148.6 ^b^	141.6, 155.5	153.1	146.4, 159.8	157.6 ^b^	150.7, 164.4	0.003
HDL-cholesterol (mg/dL)	1107	61.3	58.5, 64.0	62. 9	60.3, 65.5	63.3	60.6, 66.0	0.212
LDL-cholesterol (mg/dL)	855	75.9	71.2, 80.7	77.3	73.0., 81.6	79.6	75.1, 84.1	0.279
Non-HDL cholesterol (mg/dL)	1038	89.0 ^b^	84.4, 93.6	91.9	87.7, 96.0	95.1 ^b^	90.8, 99.5	0.027
Triglycerides (mg/dL)	1108	79.0	74.4, 83.7	78.3	74.3, 82.3	79.1	74.7, 83.5	0.942
TC:HDL ratio	956	2.56	2.47, 2.65	2.50	2.43, 2.58	2.57	2.49, 2.65	0.318
LDL:HDL ratio	782	1.33	1.25, 1.41	1.29	1.22, 1.36	1.35	1.28, 1.43	0.386

BMI: Body mass index (kg/m^2^); WC: Waist circumference; BP: Blood pressure; FMI: Fat mass index; TC: Total cholesterol. * Estimated marginal means from multilevel linear mixed-effects models. Variables in the models were age, gender, fasting status, parental education level, migrant background, z-BMI, tanner stage, and moderate and vigorous physical activity and included a random school and a random municipality effect to account for the clustered study design. ^a^ Significant differences between *Processed* and *Traditional* clusters after Bonferroni correction; ^b^ Significant differences between *Processed* and *Healthy* clusters after Bonferroni correction.

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
