# Peer review of "Dietary Patterns and Cardiovascular Risk Factors in Spanish Adolescents: A Cross-Sectional Analysis of the SI! Program for Health Promotion in Secondary Schools"

_nutrients, 2019, doi:10.3390/nu11102297_

Round 1
Reviewer 1 Report
Overall, this is a solidly conducted, important study and a well-written manuscript. The authors clearly thought through every aspect of the approach and analysis and provided a well-balanced and detailed discussion of the findings. Given prior research in other populations and some limitations of the study methodology, the findings are not surprising.
One major drawback to the analysis is that energy intake was not quantified and therefore analysis could not be adjusted for total energy intake. This might have been very important given the fact that significant socioeconomic differences were detected between dietary patterns; specifically, adolescents from families of lower socioeconomic status may resort to unhealthier food options, but might still be underconsuming total energy due to food insecurity, which in turn might explain the differences observed in clinical markers. The authors mention this limitation, but it is not explicitly discussed in the context of socioeconomic status.
Further, I am not sure what to make of the traditional dietary pattern (PCA and clusters) as it is neither healthy nor particularly unhealthy. The interpretation of the results associated with this pattern seems difficult.
Introduction, line 45-46: I suggest avoidance of broad generalizations about adolescents' intake of foods with low nutritional quality. It would have been helpful to include a comparison of what we know of adults and adolescents' dietary intake; it appears that adolescents are singled out as the one population group with particularly low nutritional quality while no information is provided on whether or not it is worse or the same compared to adult nutrition. The claim in the introduction seems oversimplified and somewhat biased. In addition, given socioeconomic differences in dietary patterns, some information could have been added to the introduction.
In line with previous research in adolescents, the main message from this study is that the differences observed in clinical markers were not clinically significant, because they were all still within normal range. I suggest that this detail, as mentioned in the discussion, is added to the conclusion of the abstract.
Author Response
Reviewer 1:
Reviewer comment 1: Overall, this is a solidly conducted, important study and a well-written manuscript. The authors clearly thought through every aspect of the approach and analysis and provided a well-balanced and detailed discussion of the findings. Given prior research in other populations and some limitations of the study methodology, the findings are not surprising.
One major drawback to the analysis is that energy intake was not quantified and therefore analysis could not be adjusted for total energy intake. This might have been very important given the fact that significant socioeconomic differences were detected between dietary patterns; specifically, adolescents from families of lower socioeconomic status may resort to unhealthier food options, but might still be underconsuming total energy due to food insecurity, which in turn might explain the differences observed in clinical markers. The authors mention this limitation, but it is not explicitly discussed in the context of socioeconomic status.
Authors’ response: We would like to thank Reviewer #1 for his/her thoughtful feedback. We agree with the reviewer about the comments done about the limitations of the methodology used in our study. The food frequency questionnaire was used as a screening tool to assess the association of eating behaviours with risk factors and cardiovascular health in children; however, it does not allow to quantify total energy intake. As the reviewer points out, this is a significant limitation of the study, especially when it comes to the study of the associations between dietary patterns and clinical markers in low socioeconomic populations. Following the reviewer’s suggestion, we have added the following paragraph in the discussion section of the revised version of the manuscript (Page 13, line 221-224):
“This might be especially relevant in the context of socioeconomic differences. In our study, adolescents from families reporting a relatively lower socioeconomic status were more frequently allocated to the Processed pattern but still could be consuming less total energy than required due to food insecurity, and thus confounding the associations with clinical markers.”
Reviewer comment 2: Further, I am not sure what to make of the traditional dietary pattern (PCA and clusters) as it is neither healthy nor particularly unhealthy. The interpretation of the results associated with this pattern seems difficult.
Authors’ response: We agree with the reviewer about his/her comments about the traditional dietary pattern. Among the three derived patterns (either PCA or clusters), the traditional one was the less specific and somehow in between the healthy and the processed patterns. Interestingly, this pattern was extracted from both PCA and Cluster analysis procedures showing very similar profiles. This profile depicts a dietary pattern which is quite common in our communities, characterized on the one hand by frequent consumption of white bread, fresh meat, processed/cured meat (especially ham), cheese, sweetened milk/ chocolate milk, pasta and water, and on the other hand by “in-between” consumption of processed products. This pattern was not the healthiest or the unhealthiest, and the associations with body composition outcomes were also “in-between”. Following the reviewer’s comment, we have added the following sentence in the discussion section of the revised manuscript (Page 13, line 242-44):
“Conversely, the lowest agreement was found for the Traditional patterns; they tended to correlate with the Processed patterns and showed weaker associations with clinical markers, making their interpretation more difficult.”
Reviewer comment 3: Introduction, line 45-46: I suggest avoidance of broad generalizations about adolescents' intake of foods with low nutritional quality. It would have been helpful to include a comparison of what we know of adults and adolescents' dietary intake; it appears that adolescents are singled out as the one population group with particularly low nutritional quality while no information is provided on whether or not it is worse or the same compared to adult nutrition. The claim in the introduction seems oversimplified and somewhat biased. In addition, given socioeconomic differences in dietary patterns, some information could have been added to the introduction.
Authors’ response: We thank the reviewer for his/her comment. Following the reviewer’s suggestion, we have deleted such claim from the introduction and briefly commented on socioeconomic differences in dietary patterns as follows (Page 2, line 45-47):
“The consumption of low nutritional quality foods is especially prevalent in adolescents from low socioeconomic status families (7,8). Nevertheless, according to the results of the HELENA study, the proportion of European adolescents achieving the AHA ideal diet score is as low as 2%.”
Reviewer comment 4: In line with previous research in adolescents, the main message from this study is that the differences observed in clinical markers were not clinically significant, because they were all still within normal range. I suggest that this detail, as mentioned in the discussion, is added to the conclusion of the abstract.
Authors’ response: We thank the reviewer for the suggestion. This information is now on the conclusion section of the abstract (Page 2, line 32-34):
“In conclusion, the overall association between DPs, as assessed by two different methods, and most analysed CVRF was weak and not clinically relevant in a large sample of adolescents”.
Reviewer 2 Report
Bodega P et al performed cluster analysis and principal component analysis to identify dietary patterns of Spanish adolescents and examined the associations between dietary patterns and cardiovascular risk factors in the same cohort. The authors provided adequate background on the significance of this study and presented the results concisely. The issues addressed in the Discussion aligned closely with the study findings. From the nutritional epidemiological point of view, comparing dietary patterns derived from two methods in the same paper was an informative approach. In the last paragraph of the Discussion section, I would suggest discussing the study strength before the study limitations.
Author Response
Reviewer 2:
Reviewer comment 1: Bodega P et al performed cluster analysis and principal component analysis to identify dietary patterns of Spanish adolescents and examined the associations between dietary patterns and cardiovascular risk factors in the same cohort. The authors provided adequate background on the significance of this study and presented the results concisely. The issues addressed in the Discussion aligned closely with the study findings. From the nutritional epidemiological point of view, comparing dietary patterns derived from two methods in the same paper was an informative approach. In the last paragraph of the Discussion section, I would suggest discussing the study strength before the study limitations.
Authors’ response: We would like to thank the reviewer for the consideration given to our work. Following the reviewer’s suggestion, we have reordered the last part of the discussion section as follows (Page 14, line 252-262):
“This study has several strengths. The SI! Program for Secondary School is a large multi-centre study in which all nurses and nutritionists received prior training according to strictly standardized procedures for data collection and guiding participants in completing the questionnaire. Another strength is the use and comparison of two different methods to derive DPs, thus providing more robust evidence and consistency in the results found. Moreover, the study focuses on pre-adolescents, who receive less attention in lifestyle studies than younger children or older adolescents. Finally, follow-up of these adolescents will provide the opportunity to track changes in DPs and to assess their association with changes in CV health over time. Some study limitations should be acknowledged. The cross-sectional nature of the analyses allows the establishment of associations but not causation. Participation in the study was voluntary, and we therefore cannot exclude selection bias due to adolescents with worse CV health or unhealthier lifestyles being less willing to participate. Another limitation is the self-reported nature of the CEHQ-FFQ, which introduces the possibility of social desirability bias or difficulties in the accurate estimation of intake frequency for certain food groups”.